# Micronutrient-deficient diets and possible environmental enteric dysfunction in Buruli ulcer endemic communities in Ghana: Lower dietary diversity and reduced serum zinc and vitamin C implicate micronutrient status a possible susceptibility factor

**Aloysius Dzigbordi Loglo**[1,2], **Philemon Boasiako Antwi**[2], **Kabiru Mohammed Abass**[3], **Samuel Osei-Mireku**[3], **George Amofa**[4], **Elizabeth Ofori**[5], **Jonathan Kofi Adjei**[2], **Michael Ntiamoah Oppong**[2], **Richard Odame Phillips**[2,6], **Reginald Annan**[7], **Barbara Engel**[8], **Rachel E. Simmonds**[1]*

1 Microbes, Infection & Immunity, School of Biosciences, Faculty of Health and Medical Sciences, University of Surrey, Guildford, United Kingdom, 2 Kumasi Centre for Collaborative Research in Tropical Medicine (KCCR), Kwame Nkrumah University of Science and Technology, Kumasi, Ghana, 3 Agogo Presbyterian Hospital, Agogo, Ghana, 4 Dunkwa Government Hospital, Dunkwa, Ghana, 5 Tepa Government Hospital, Tepa, Ghana, 6 Department of Medicine, School of Medical Sciences, Kwame Nkrumah University of Science and Technology, Kumasi, Ghana, 7 Department of Biochemistry and Biotechnology, College of Sciences, Kwame Nkrumah University of Science and Technology, Kumasi, Ghana, 8 Department of Nutritional Sciences, School of Biosciences, University of Surrey, Guildford, United Kingdom

* rachel.simmonds@surrey.ac.uk

## Abstract

### Background

The nutritional status of communities susceptible to Buruli ulcer (BU, a skin NTD caused by infection with *Mycobacterium ulcerans*) remains almost completely obscure. We have assessed the diets of BU patients vs. controls from the same BU-endemic communities, and compared their circulating biomarkers of nutrients and inflammation.

### Methods/Principal Findings

We investigated two cohorts of BU patients and controls. The first were administered food frequency and multi-pass 24-hour recall questionnaires to determine patterns of foods consumed, nutrient intake and nutrient adequacy. The second used archived serum samples collected as baseline to measure the circulating concentration of zinc, vitamin C, CRP, IL-1β, IFN-γ, TNF-α and IL-6. Stunted growth was more prevalent than expected (31%), while 18% of participants were underweight and most had inadequate intake of all micronutrients except for carbohydrate. BU patients had a lower intake of, selenium, vitamin B₁₂ and zinc, and for selenium and vitamin B₁₂ a higher proportion had dietary insufficiency (40% vs. 15% and 80% vs. 55%, respectively). In line with this, BU patients

**Data availability statement:** The authors confirm that all data underlying the findings are fully available without restriction. All relevant data are within the paper and its Supporting Information files.

**Funding:** This work was supported by the Wellcome Trust (WT202843_Z_16 to RES), Nestlé Foundation (to ADL, ROP and RA) and the University of Surrey (to ADL and RES). The funders had no role in study design, data collection and analysis, decision to publish, or preparation of the manuscript.

**Competing interests:** The authors have declared that no competing interests exist.

had significantly lower levels of zinc in their serum, and more had levels below the normal range (72% vs. 43%). Despite many participants having a good intake of vitamin C, serum levels were low, and lower amongst the BU patients. As expected, there was little evidence of systemic inflammation (CRP <0.6 mg/L). Elevated IL-6 levels were present in several participants suggesting that environmental enteric dysfunction may be prevalent in these communities, however this was similar in cases vs. controls.

## Conclusions/Significance

Diet and nutritional status may be a contributing factor to BU pathogenesis. Protein and the micronutrients zinc, selenium, vitamin $B_{12}$ and vitamin C may be of particular importance. Nutritional interventions may have potential for both prophylaxis and treatment of BU, which may be a cost-effective approach to achieving the NTD Roadmap goals.

### Author summary

Buruli ulcer is a serious and debilitating neglected tropical disease of public health concern, one of several with skin manifestations known as Skin NTDs. Buruli ulcer mostly affects rural communities in some parts of sub-Saharan Africa. The diets of these communities are rarely studied, including potential links with infectious diseases. In this paper, we employed dietary surveys and blood analysis to investigate the potential links between nutrition and Buruli ulcer in two sets of patients and controls from the Ashanti and Central regions of Ghana. We found that the diets of the study population were inadequate in many vitamins and minerals. Their blood levels of vitamin C and zinc were also frequently lower than normal. The Buruli ulcer patients had poorer diets and were also more likely to have vitamin and mineral inadequacies compared to matched controls. Supplementing the diets of at-risk communities and patients with Buruli ulcer may therefore be valuable in its prevention and treatment, respectively. It could also be a cost-effective strategy to reduce the burden of skin NTDs more broadly.

## Introduction

Buruli ulcer (BU) is a necrotizing skin disease caused by the non-tuberculous *Mycobacterium ulcerans* [1]. It is one of the neglected tropical diseases (NTDs) with skin manifestations [2], occurring as a painless nodule, plaque, oedema or ulcer [3]. A pathogen-derived exotoxin, mycolactone, is known to play a central role in disease pathogenesis [4,5] due to its blockade the Sec61 translocon [6] underpinning its cytotoxic and immunosuppressive activities [7] as well as its induction of vascular dysfunction [8]. Hence, patients with BU do not have a systemic inflammatory response and generally present without fever or malaise even when ulceration is extensive [9,10]. However, it is known that the immune system does control susceptibility to BU, with evidence from patients and animal models showing that macrophage function can influence infection outcomes [11–13].

Globally, BU is the third most prevalent mycobacterial disease and is commonly found in tropical or semitropical climates and most frequently in West Africa and Australia [1,10,14]. In Africa, the disease is widespread in poor rural communities with most cases found in children below 15 years. Even in high-resource settings, delays in treatment can result in

permanent skin deformation or lifelong disabilities. Early case detection is currently the most effective approach for harm-minimisation as no vaccine exists.

To seek novel and cost-effective prevention approaches, and knowing that nutritional inadequacy is a risk factor for other mycobacterial infections (for example tuberculosis [15,16] and leprosy [17,18]) because it compromises immunity [19,20], we postulated that this may also be the case for BU. However, there is a paucity of data for both BU patients and marginalised rural subsistence communities in West African countries. We, therefore, aimed to fill this knowledge gap by investigating the potential role of nutrition in the pathogenesis of BU.

## Methods

### Ethics statement

This study received favourable ethical opinion from the School of Medical Sciences Committee on Human Research, Publication and Ethics of KNUST (CHRPE/AP/18, CHRPE/AP/310/21) and the Faculty of Health and Medical Sciences Ethics Committee of the University of Surrey (UEC 2017 98 FHMS). All adult study participants gave written informed consent. Written formal consent was obtained from the parent/guardian for child participants.

### Study area

BU patients were recruited from Agogo Presbyterian Hospital, Dunkwa Government Hospital, and Tepa Government Hospital in Ghana, which are located in areas of known BU endemicity. The archived serum samples were from previously published cohorts [21,22].

### Study design, population, and sampling

BU patients <5 years old, pregnant and lactating mothers were excluded from the study. Demographic and clinical data of cases was collected on a standardised skin NTDs [23] or BU01 form [21,22]. The type and category of the lesion were assigned based on the WHO case definition [24]. Ulcerated lesions were dressed by nurses according to the WHO-recommended wound management protocol [24]. BU case confirmation was by a validated combined 16S rRNA reverse transcriptase/IS2404 qPCR assay [22] or IS2404 qPCR [21,22].

Controls were recruited from the same study sites to ensure an unbiased reflection of the socioeconomic status, and food availability of the study population. Controls were age-sex-matched (as closely as possible) and residing in the same community of a recruited BU patient but from a different household.

### Anthropometry

Anthropometric measures were obtained from as many participants as possible. Stratification of BMI was as advised by the WHO [25] including BMI-for-age estimation for participants 5-19 years [26,27]. Anthropometry data had not been collected for the controls in Cohort 2. The height-for-age z-score (HAZ) was used to determine the stunting status of young people using AnthroPlus (WHO; Version 1.0.4.0) [27,28].

### Dietary assessment (Cohort 1)

A Ghana-specific food frequency questionnaire (FFQ) made up of 105 foods (S1 Table) was used to assess the frequency with which food items were consumed over the previous month at the time of recruitment (daily, weekly, monthly, and rarely/never) [29,30]. These foods were divided into 25 food groups, using an approach adapted from a previous study of Ghanaian diets [31].

A Ghana-specific multi-pass 24-hour dietary recall questionnaire was also administered to obtain information on foods consumed for two week-days and one weekend day [32]. Household measures and food models were used to aid easy quantification of foods consumed, which were then converted into grams using a standardized unit of measure provided on a nutrient analysis template for each participant, which also estimated the nutritional content of these foods [33, 34]. Each participant's nutrient intake was compared to the relevant cut-off for their age (and gender if relevant), using information from the Estimated Average Requirement (EAR) of Food and Agriculture Organization (FAO)/WHO [35] or the Recommended Dietary Allowance (RDA) [36] if no EAR was available. The dietary diversity score (DDS) was also generated from the 24-hour recall data. This had 15 food groups (S2 Table) adapted from FAO guidelines for measuring household and individual dietary diversity [37], but with different numbers of food groups.

We assessed potential socioeconomic factors influencing the diets of participants using a questionnaire to obtain data on the gender, household income, education level and occupation of study participants. The household income was estimated based on the reported household monthly income per individual. Some of the participants depended on foodstuffs from their farms or livestock rearing. Those participants made an estimate on the current cost of the foods consumed from their farms or livestock to help ensure accurate determination of the monthly household income. Parents were allowed to make the estimation of household income for their children.

The highest educational status of the participants was categorized into elementary (none, primary or junior-high), secondary (senior-high and post-secondary) and higher (tertiary education only) [38]. The occupation was grouped into formal or informal. Formal included professionals, managers, clerical support staff, higher grade routine non-manual employees service and sales-related occupations while informal consisted of craft and related trades workers, elementary occupations (work that requires unskilled and semi-skilled labour) and farmers [38].

## Quantification of nutritional and inflammatory biomarkers (Cohort 2)

Vitamin C, zinc, and inflammatory markers were assessed in archived serum samples had been stored at stable -80°C for up to 8 years. Vitamin C and CRP were assessed by competitive and sandwich ELISA, respectively (MBS2700398, MBS8807458 MyBioSource, San Diego, USA). Zinc quantification used a colorimetric assay (MBS841453 MyBioSource).

A bespoke Cytometric Bead Array (CBA) LEGENDplex (Biolegend, USA) was used for the simultaneous detection of four cytokines [IL-1β (B326518), IL-6 (B326531), IFN-γ (B326521) and TNF-α (B326522)] in 92 serum samples according to the manufacturer's instructions.

Vitamin C and zinc concentrations were compared to normal ranges that have been determined in and for European populations [39–42]. We have no reason to believe that the nutritional requirements for Africans would vary widely from these values.

## Statistical analysis

Data were analysed using GraphPad Prism version 9.4.1 (GraphPad Software Inc., USA). All continuous variables were tested for normal distribution. Non-Gaussian data compared medians between two groups using a one- or two-tailed Mann-Whitney-U test. Multiple comparisons used the Kruskal-Wallis test together with the Dunn`s post-hoc correction. Multiple t-tests were used to compare nutrient intake between patients and controls, as many nutrients correlated with each other (determined in Pearson correlation matrix). To compare the frequency of categorical variables we used Fisher's exact test or Chi-squared where there were more than two variables.

## Results

### Characteristics of the study populations

We recruited 40 confirmed BU cases and 40 controls in Cohort 1, and identified archived samples from 80 confirmed BU cases and 80 controls in Cohort 2 (Table 1). The median age of the cohorts was 19.5 (IQR 11-34) and 14.5 (IQR 10-27) years, respectively (range 5-63 years). In both cohorts the majority of cases presented with ulcers, and lesions were predominately of Category I or II (80% and 73% respectively). Notably the rate of category III lesions was higher in the most recently recruited patients (Cohort 1), probably due to the disruption in case detection due to the COVID-19 pandemic. In line with the age profile of Cohort 1, the approximately half had (or were having) primary school education, and the socioeconomic profile was similar between cases and controls (S3 Table).

While the majority of the participants had normal weight categorisation, 18.1% were underweight including 22.5% and 19% of BU cases and 12.5% of controls. In Cohort 1 the difference in proportion of underweight participants was not statistically significant (Table 1). Among the 86 young participants for whom we had anthropometry data, 27 (31.4%) had stunted growth (Table 1). The relatively low numbers in the subgroups make it hard to compare BU cases and controls, but in Cohort 2, 3 out of 46 BU cases (7%) were categorised as being severely stunted.

### Frequency of food consumption

The FFQ gave qualitative insights into the frequency with which Cohort 1 study participants ate foods within a range of different food groups (S1 Fig). Overall, this analysis shows that the participants in this study ate a fairly typical rural Ghanaian diet consisting of carbohydrate-rich foods. For example, whole grain cereals, refined cereals, fermented maize products, rice and pasta, roots/tubers/plantain were eaten frequently by more than half the participants (S1A–S1E Fig). This was alongside protein sources (predominantly chicken, fish and legumes S1F, S1M and S1L Fig), fruit (S1J Fig) and vegetables (S1G–S1I Fig). Some participants also partook in "Western" processed foods and drinks such as sodas and sweets/cakes (S1O and S1Q Fig), although more than three-quarters of the participants reported they never ate potatoes or processed meat (S1R–S1Q Fig). However, since the FFQ did not include a measure of quantity, it was hard to know how fully this meets the dietary requirements of the participants.

### Nutrient intake and dietary diversity

A multi-pass 24-hour recall questionnaire was used to determine the overall energy, as well as dietary macro- and micro-nutrient, intake of Cohort 1 study participants. This showed that, on average, the proportion of macronutrients making up the daily intake was close to expected norms (S4 Table). In line with the FFQ, carbohydrate was the predominant nutrient in the meals of both BU cases and controls (64% of daily energy intake), which is at the higher end of expected norms (45-65%). On the other hand, protein made up the smallest portion of energy intake (9.6%), at the lower end of the normal requirement of 10-35%.

While nutrient distribution was relatively normal, the energy intake (1629±638 Kcal, Table 2) for the vast majority of the participants was inadequate, with only 15% meeting this requirement (Table 3). For macronutrients, while almost all Cohort 1 participants had adequate dietary carbohydrate, only 49% of the study population had inadequate intake of protein (Table 3). There were no differences in total daily macronutrient intake between BU cases and controls (Table 2).

**Table 1. Characteristics of study participants.**

| Variables | Cohort 1 | | | Cohort 2 | | |
|---|---|---|---|---|---|---|
| | BU cases | Controls | P-value | BU cases | Controls | P-value |
| | N=40 | N=40 | | N=80 | N=80 | |
| **Age range (years)** | 5-63 | 5-58 | 0.530[a] | 5-56 | 5-62 | 0.9599[a] |
| *Median (IQR)* | 17 (10-38) | 21.5 (13-29) | | 14.5 (10-25) | 14.5 (10-28) | |
| **Gender, N (%)** | | | | | | |
| *Male* | 19 (48%) | 22 (55%) | 0.654[b] | 39 (49) | 40 (50) | >0.9999[b] |
| *Female* | 21 (52%) | 18 (45%) | | 41 (51) | 40 (50) | |
| **Lesion type, N (%)** | | | | | | |
| *Nodule* | 3 (8%) | – | | 19 (24%) | – | |
| *Plaque* | 11 (28%) | – | | 18 (23%) | – | |
| *Oedema* | 0 (0%) | – | | 1 (1%) | – | |
| *Ulcer* | 25 (63%) | – | | 42 (53%) | – | |
| *Osteomyelitis* | 1 (3%) | – | | 0 (0%) | – | |
| **Lesion Category, N (%)** | | | | | | |
| *I* | 14 (35%) | – | | 42 (53%) | – | |
| *II* | 18 (45%) | – | | 32 (40%) | – | |
| *III* | 8 (20%) | – | | 6 (8%) | – | |
| **Waist-hip ratio[d]** | | | | | | |
| Median | 0.88 | 0.90 | 0.643[a] | | | |
| IQR | 0.73-1.02 | 0.77-0.96 | | | | |
| **Weight categorisation[d,e], N (%)** | | | | | | |
| All | 40 | 40 | | 69 | | |
| *Underweight* | 9 (22.5%) | 5 (12.5%) | 0.582[c] | 13 (19%) | N/A | |
| *Normal* | 27 (67.5%) | 31 (77.5%) | | 51 (74%) | N/A | |
| *Overweight* | 3 (7.5%) | 2 (5%) | | 5 (7%) | N/A | |
| *Obese* | 1 (2.5%) | 2 (5%) | | | | |
| Adult (≥20yr) | 18 | 20 | | 23 | | |
| *Underweight* | 3 (16.7%) | 3 (15%) | | 0 (0%) | N/A | |
| *Normal* | 13 (72.2%) | 15 (75%) | | 19 (83%) | N/A | |
| *Overweight* | 1 (5.6%) | 1 (5%) | | 4 (17%) | N/A | |
| *Obese* | 1 (5.6%) | 1 (5%) | | | | |
| Young people (5-19yr) | 22 | 18 | | 46 | | |
| *Underweight* | 6 (27.3%) | 1 (5.6%) | | 13 (28%) | N/A | |
| *Normal* | 14 (77.8%) | 15 (83.3%) | | 32 (70%) | N/A | |
| *Overweight* | 2 (11.1%) | 1 (5.6%) | | 1 (2%) | N/A | |
| *Obese* | 0 (0%) | 1 (5.6%) | | | | |
| **Height-for-age status (5-19yr)[d,f], N (%)** | | | | | | |
| *Not stunted* | 15 (68.2%) | 14 (77.8%) | 0.734[b] | 30 (65%) | N/A | |
| *Moderately stunted* | 7 (31.8%) | 4 (22.2%) | | 13 (28%) | N/A | |
| *Severely stunted* | 0 (0%) | 0 (0%) | | 3 (7%) | | |

Counts of demographic, clinical, and socioeconomic characteristics of study participants. [a]; Two-tailed Mann-Whitney tests, [b]; Fisher's exact test, [c]; $\chi^2$ test (p-value < 0.05 was considered statistically significant), [d]; Anthropometric data was available for 69 BU patients and no controls in Cohort 2, [e]; Weight categorisation used BMI-for-age for study participants aged 5-19 [31], and BMI for adults, [f]; stunting according to WHO AnthroPlus. Abbreviation: SD; Standard Deviation, IQR; Interquartile range, N/A: not available.

**Table 2. Comparison of nutrient intake amounts between Cohort 1 participants.**

| Nutrients | All | BU Cases | Controls | P-value |
|---|---|---|---|---|
| | N=80 | N=40 | N=40 | |
| | *Mean± SD* | *Mean± SD* | *Mean± SD* | |
| Energy (Kcal) | 1629±638 | 1594±625.7 | 1664±655.9 | 0.624 |
| Carbohydrate (g) | 292±259 | 261.9±119.1 | 322.7±345.7 | 0.298 |
| Protein (g) | 42.2±19.6 | 38.84±16.32 | 45.63±22.05 | 0.122 |
| Fat (g) | 47.3±23.7 | 46.99±27.29 | 47.52±19.88 | 0.921 |
| Fibre (g) | 23.5±10.7 | 23.72±10.18 | 23.23±11.29 | 0.838 |
| Folate (µg) | 297±176 | 303.3±181.4 | 289.8±172.6 | 0.733 |
| Iron (mg) | 9.94±4.68 | 9.58±4.60 | 10.29±4.80 | 0.500 |
| Selenium (µg) | 65.00±40.5 | 52.69±31.71 | 77.35±44.53 | **0.006** |
| Vitamin A (µg) | 597±715 | 573.2±645.7 | 620.0±785.8 | 0.773 |
| Vitamin $B_{12}$ (µg) | 1.61±2.11 | 0.89±0.92 | 2.332±2.68 | **0.002** |
| Vitamin C (mg) | 110±77.0 | 124.8±91.40 | 95.01±56.54 | 0.084 |
| Vitamin E (mg) | 6.22±3.43 | 6.096±3.961 | 6.340±2.86 | 0.752 |
| Vitamin K (µg) | 49.4±44.5 | 48.88±37.36 | 50.94±51.14 | 0.761 |
| Zinc (mg) | 6.23±2.49 | 5.55±1.70 | 6.90±2.96 | **0.015** |

Data were compared between cases and controls using an unpaired t-test. P-value <0.05 are indicated in bold text.

Nutritional adequacy was determined using the EAR for all nutrients except for energy, fibre and vitamin K that do not have an EAR, and here RDA was used instead. Both vary depending on age, gender, height, pregnancy, and lactation (although pregnant and lactating mothers were excluded from the study). [a] two-sided Fisher's exact test. P-value <0.05 are indicated in bold text.

In terms of the micronutrient intake of Cohort 1, while the majority of the cohort had adequate intakes of dietary iron, selenium, and vitamin C (74%, 72%, and 82%, respectively, Table 3), >60% had inadequate intakes of dietary vitamin A (60%), vitamin $B_{12}$ (68%), vitamin E (86%), vitamin K (79%), and zinc (68%). For most micronutrients, there was no statistically significant difference in nutritional adequacy between BU cases and controls. However, a higher proportion of BU patients were below the EAR for dietary selenium and vitamin $B_{12}$ compared with the controls (Table 3). Moreover, when comparing the mean intake of micronutrients calculated from the 24-hour recall interviews, selenium, vitamin $B_{12}$, and zinc were significantly lower in BU cases than controls (Table 2), and vitamin C was different with a P-value of 0.084.

There were no statistical differences in intake amongst those with different categories of BU lesion (S2 Fig), sex (S5 Table, except for folate in controls; p=0.018), or household income (S6 Table), although in some cases, this may have been driven by the power of our study.

A diverse diet is strongly associated with diet quality; therefore, we generated a DDS for the participants. This showed the diversity of the foods consumed among 15 food groups over a period of 3 days. Overall, the DDS of this cohort were within acceptable ranges for both groups, with only 6 BU cases and 4 controls consuming foods from less than 5 of these food groups in this time period (Fig 1). However, the DSS for BU cases was significantly lower than in the controls (6.2 vs. 7.2, p = 0.0320).

## Nutritional biomarkers

Based on these dietary data, we used archived serum samples to carry out a proof of principle that these dietary differences would translate to changes in serum concentrations, and

**Table 3. Percentage adequacy of nutrient intake in Cohort 1 participants.**

| Nutrients | Frequency | All | BU cases | Controls | P-value[a] |
|---|---|---|---|---|---|
| | | N=80 | N=40 | N=40 | |
| Energy (Kcal) | Adequate | 12 (15%) | 6 (15%) | 6 (15%) | 1.000 |
| | Inadequate | 68 (85%) | 34 (85%) | 34 (85%) | |
| Carbohydrate (g) | Adequate | 79 (99%) | 40 (100%) | 39 (98%) | 1.000 |
| | Inadequate | 1 (1%) | 0 | 1 (2%) | |
| Protein (g) | Adequate | 41 (51%) | 18 (45%) | 23 (57%) | 0.371 |
| | Inadequate | 39 (49%) | 22 (55%) | 17 (43%) | |
| Fibre (g) | Adequate | 28 (35%) | 17 (43%) | 11 (28%) | 0.241 |
| | Inadequate | 52 (65%) | 23 (58%) | 29 (73%) | |
| Folate (µg) | Adequate | 40 (50%) | 21 (52%) | 19 (47%) | 0.823 |
| | Inadequate | 40 (50%) | 19 (48%) | 21 (53%) | |
| Iron (mg) | Adequate | 59 (74%) | 29 (72%) | 30 (75%) | 1.000 |
| | Inadequate | 21 (26%) | 11(14%) | 10 (25%) | |
| Selenium (µg) | Adequate | 58 (72%) | 24 (60%) | 34 (85%) | **0.023** |
| | Inadequate | 22 (28%) | 16 (40%) | 6 (15%) | |
| Vitamin A (µg) | Adequate | 32 (40%) | 18 (45%) | 14 (35%) | 0.494 |
| | Inadequate | 48 (60%) | 22 (55%) | 26 (65%) | |
| Vitamin $B_{12}$ (µg) | Adequate | 26 (32%) | 8 (20%) | 18 (45%) | **0.031** |
| | Inadequate | 54 (68%) | 32 (80%) | 22 (55%) | |
| Vitamin C (mg) | Adequate | 66 (82%) | 36 (90%) | 30 (70%) | 0.139 |
| | Inadequate | 14 (18%) | 4 (10%) | 10 (25%) | |
| Vitamin E (mg) | Adequate | 11 (14%) | 8 (20%) | 3 (7%) | 0.193 |
| | Inadequate | 69 (86%) | 32 (80%) | 37 (93%) | |
| Vitamin K (µg) | Adequate | 17 (21%) | 10 (25%) | 7 (18%) | 0.586 |
| | Inadequate | 63 (79%) | 30 (75%) | 33 (83%) | |
| Zinc (mg) | Adequate | 26 (32%) | 12 (30%) | 14 (35%) | 0.812 |
| | Inadequate | 54 (68%) | 28 (70%) | 26 (65%) | |

selected zinc and vitamin C for analysis. The serum concentration of zinc and vitamin C was significantly lower in BU cases compared to controls (Fig 2A). Notably many BU patients and controls had levels below the expected normal ranges in serum, and this was significantly more frequent amongst BU patients than controls (P< 0.001, Table 4).

Severity of BU disease can be determined both by the different clinical presentations, and by the lesion category which is based on the longest diameter of the lesion (Fig 2B and 2C). There were no differences in the serum concentration of zinc or vitamin C between BU cases with less severe disease as defined by WHO category I, II or III lesion (Fig 2B), although a trend for lower vitamin C with increasing seriousness may have been limited by the few cases of Category III lesions in this cohort (n=6; 8%). However, BU patients with plaque forms had significantly lower levels of vitamin C compared to those who presented with nodules (Fig 2C), which are normally the first sign of infection.

## Inflammatory biomarkers

The serum levels of CRP, IL-1β, IFN-γ, TNF-α and IL-6 were also assessed in a subset of Cohort 2. However, with the exception of IL-6, many samples had circulating levels below the detection limit of the assays. Therefore, we took an approach that assigned each participant within a threshold (Table 5). No participants had levels of CRP that indicated the presence

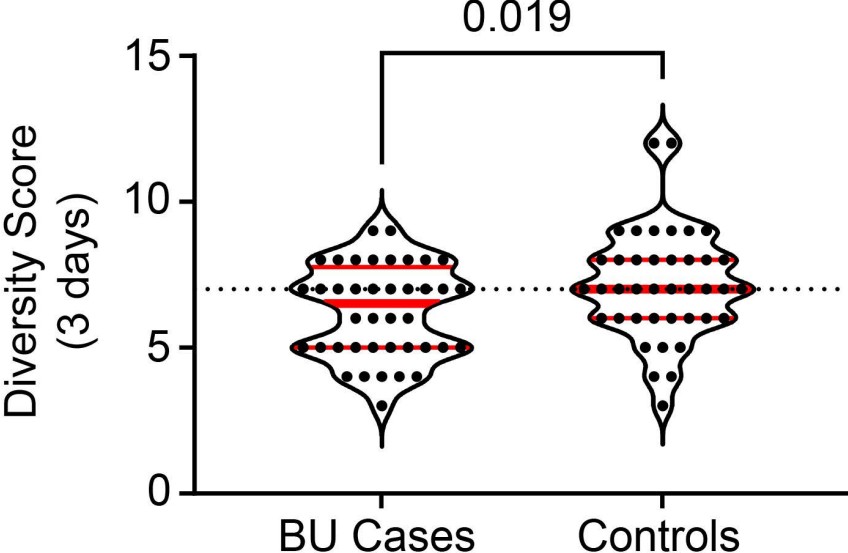

**Fig 1. Comparison of dietary diversity between BU cases and controls.** The dietary diversity score (DDS) was generated from 24-hour recall data obtained from all 80 Cohort 1 participants (40 BU cases, 40 controls), which assessed whether each participant ate any food from a particular group, out of 15, over the three days of the multi-pass (2 weekdays and one weekend day). The food groups for the were cereals, roots & tubers, green leafy vegetables, other vegetables, fruits, meat, fish & seafood, eggs, pulses & nuts, milk & milk products, oils & fats, sugar, condiments, drinks & beverages, and 'other'. Dietary diversity between BU cases and controls was compared using a two-tailed Mann-Whitney test. A p-value ≤ 0.05 was considered statistically significant.

of even "high-normal" [43] levels of CRP. For IL-1β, we chose a cut-off of 57pg/ml, and only 7-13% of Cohort 2, respectively, were above this level, (Table 5). For TNF-α and IFN-γ we choose two cut-offs, with 2-5% having levels above the higher cut-off. There was no statistically significant difference between BU cases and controls for these parameters.

For IL-6, such cut-offs are less well-defined than for CRP, but better defined than for other cytokines, including by systematic review that showed the range of IL-6 measured in cohorts of healthy patients was from 0-43.5pg/ml [44], although other reports from the global North report at the lower end of this (2-3pg/mL [43,45,46]). Hence, for our analysis, we decided to stratify our data based on whether they were normal (<7pg/ml), slightly (7-50pg/ml), moderately (50-500pg/ml) or highly (>500pg/ml) elevated (Table 5). The proportion of BU cases and controls in these different criteria were not statistically different, and neither was there a difference in serum IL-6 levels (Fig 3A). There was no significant increase in IL-6 levels within the presentation of BU (Fig 3B), nor the category of lesion (Fig 3C). However, like vitamin C, this may be due to the small number of category III lesions.

However, this analysis showed that more than half of both groups had evidence of elevated IL-6 compared to literature reports in European populations (Table 5). In order to understand if these elevated levels of IL-6 might explain the reduced zinc or vitamin C with IL-6, these were correlated (S3 Fig). No significant associations were identified for BU patients (S3A Fig) or controls (S3B Fig).

Inflammation markers serum level concentration of study participants grouped according to thresholds that indicate the degree to which the marker was elevated. Numbers in each category were compared using contingency tables, using Fisher's exact or chi-square tests with P < 0.05 considered statistically significant. [a] Includes 67 patients and 72 controls whose CRP levels were lower than the detection limit of the assay. [b] Includes 32 patients and 36 controls

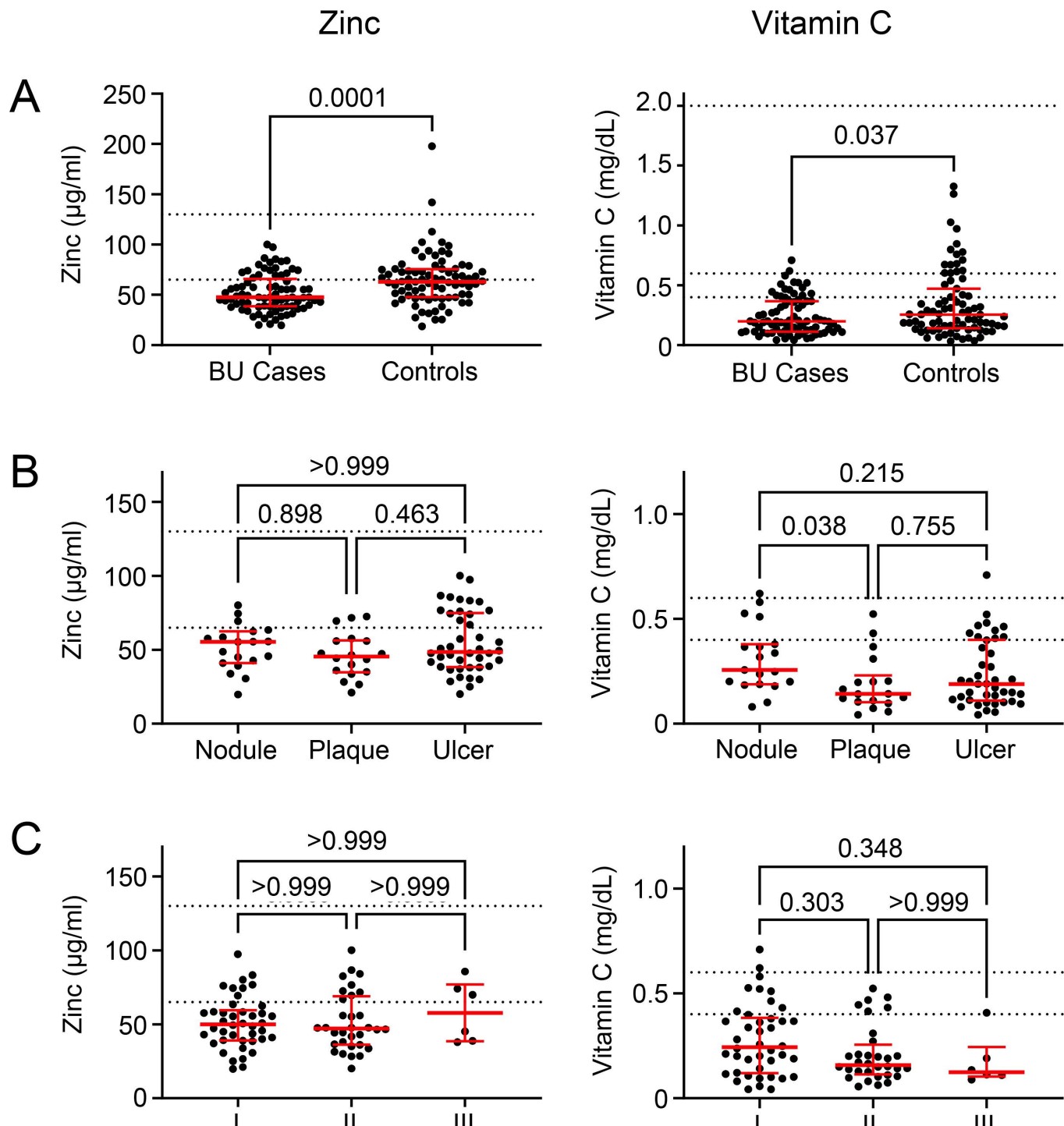

**Fig 2. BU patients have significantly lower zinc and vitamin C serum levels than controls.** The serum concentrations of zinc and vitamin C were assessed in BU cases and controls from Cohort 2 (40 BU cases, 40 controls). Individual data representing the average of duplicate assays is displayed along with the median and interquartile range. The expected normal ranges for each analyte are indicated with the dotted line (zinc 65-130μg/dL; vitamin **C** 0.4/0.6-2 mg/ dL). P-values are displayed, with **P** < 0.05 considered significant. A. Data were compared using a one-tailed Mann-Whitney test. B&C. Subgroup analysis for BU cases with different clinical forms of the disease (N = 19, 18, and 42 for nodule, plaque and ulcer respectively, B) and different WHO lesion categories at presentation (N = 42, 32, and 6 for category I, category II, and category III lesions respectively, C). Data were compared between groups using the Kruskal-Wallis test and Dunn post-hoc test for non-parametric data.

**Table 4. Serum zinc and vitamin C below the normal range are more common in BU patients than controls in Cohort 2.**

| Markers | Description | Threshold | All, N (%) | BU Cases, N (%) | Controls, N (%) | P-value[a] |
|---|---|---|---|---|---|---|
| | | | | N=80 | N=80 | |
| Zinc | Normal | >60 μg/dL | 68 (42%) | 22 (28%) | 46 (57%) | 0.0002 |
| | Reduced | ≤60 μg/dL | 92 (58%) | 58 (72%) | 34 (43%) | |
| Vitamin C | Normal | >0.6 mg/dL | 19 (12%) | 2 (3%) | 17 (21%) | 0.0003 |
| | Reduced | ≤0.6 mg/dL | 141 (88%) | 78 (97%) | 63 (79%) | |
| | Normal | >0.4 mg/dL | 72 (45%) | 16 (20%) | 56 (70%) | <0.0001 |
| | Reduced | ≤0.4 mg/dL | 88 (55%) | 64 (80%) | 24 (30%) | |

[a]Fisher's exact test with P < 0.05 considered statistically significant.

**Table 5. Comparable serum concentration of inflammation markers in patients and controls in Cohort 2.**

| Markers | Description | Threshold | BU Cases, N (%) | Controls, N (%) | P-value |
|---|---|---|---|---|---|
| Total | | | 45 | 47 | |
| hs-CRP | Normal | <0.6 mg/L[a] | 80 (100) | 80 (100) | ND |
| | High normal | 0.6-3 mg/L | 0 (0) | 0 (0) | |
| | Minor | 3-10 mg/L | 0 (0) | 0 (0) | |
| | Moderate | 10-80 mg/L | 0 (0) | 0 (0) | |
| | Major | >80 mg/L | 0 (0) | 0 (0) | |
| IL-1β | Normal/slightly elevated | ≤57 pg/mL[b] | 42 (93) | 41 (87) | 0.4861 |
| | Elevated | >57 pg/mL | 3 (7) | 6 (13) | |
| TNF-α | Normal | <7 pg/mL[c] | 37 (82) | 36 (77) | 0.6094[e] |
| | Slightly elevated | 7-50 pg/mL | 6 (13) | 9 (19) | |
| | Elevated | >50 pg/mL | 2 (5) | 2 (4) | |
| IFN-γ | Normal | <5 pg/mL[d] | 32 (71) | 34 (72) | >0.9999[e] |
| | Slightly elevated | 5-50 pg/mL | 12 (27) | 12 (26) | |
| | Elevated | >50 pg/mL | 1 (2) | 1 (2) | |
| IL-6 | Normal | <7 pg/mL | 21 (47) | 22 (49) | 0.8645[f] |
| | Slightly elevated | 7-50 pg/mL | 15 (33) | 16 (36) | |
| | Elevated | 50-500 pg/mL | 6 (13) | 6 (13) | |
| | Highly elevated | >500 pg/mL | 3 (7) | 1 (2) | |

whose IL-1β levels were lower than the detection limit of the assay. [c] Includes 28 patients and 30 controls whose TNFα levels were lower than the detection limit of the assay. [d] Includes 18 patients and 24 controls whose IFNγ levels were lower than the detection limit of the assay. [e] Participants in slightly elevated and elevated categories had to be combined for this analysis. [f] Elevated and highly elevated categories had to be combined for this analysis.

## Discussion

The intimate links between diet, the immune system, and infectious disease have been known for some time [47]. Indeed, malnutrition has been shown to be a predominant underlying cause of death from a range of infectious diseases [48,49]. Inadequate dietary intake results in decreased immune function which leads to increased incidence of infection/disease and consequently increased severity and duration of disease as the systemic and local responses to infection are impaired. In addition, many infectious diseases lead to decreased intake or malabsorption of, or increased requirements for, proteins (amino acids), specific fatty acids,

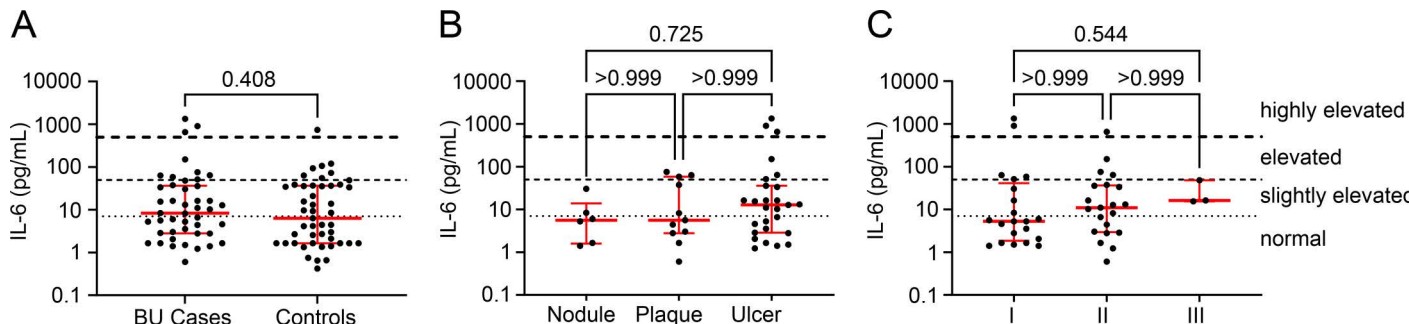

**Fig 3. IL-6 levels are not significantly different between BU patients vs controls, between BU lesion forms or lesion category.** Serum IL-6 was quantified in 45 BU patients and 47 controls and presented on a $\log_{10}$ scale. A. Data for patients and controls were compared using a two-tailed Mann-Whitney test. B. Levels were compared between BU patients with different clinical presentations of the disease (Nodules, N=6; Plaques; N=11 and Ulcer; N=27), using the Kruskal-Wallis test with Dunn's correction for multiple comparisons. C. Levels were compared between BU patients with different WHO lesion category at presentation (Category I; N=21, Category II; N=21 and Category III; N=3), using the Kruskal-Wallis test with Dunn's correction for multiple comparisons. GraphPad Prism version 9.4.1 (GraphPad Software Inc., USA) was used for all analyses. P values are displayed, with **P** < 0.05 considered significant.

vitamins and trace elements. The increased metabolic demand for specific nutrients can also lead to their deficiency leading to a vicious cycle of the emergence of new infections or resurgence of old infectious diseases.

Perhaps because of the perceived dominant effect of the cytotoxin mycolactone in disease progression, the role of diet in BU has received little attention to date. In the current work, we show that many of those living in BU endemic regions of Ghana have diets inadequate in all macronutrients except carbohydrates, as well as vitamins A, B$_{12}$, E and K, and zinc. Moreover, BU patients' diets were worse than their control counterparts who had similar age and socioeconomic backgrounds and who were also from the same villages as the BU patients (and therefore had access to similar foods). Our approach to nutritional intake used two independent tools, both of which have been used previously in rural and urban Ghanaian settings [32,50]. These were appropriate for low-literacy populations which was the case here, and for those who reside in rural and at times remote communities.

## Insights into the diet of BU endemic areas

The WHO defines malnutrition as the deficiency or excess of nutrient intake, an imbalance of nutrients or impaired nutrient utilisation in children and adults. Undernutrition can manifest in different ways including wasting, stunting, underweight and micronutrient deficiencies. On the other hand, excessive nutrients can cause overweight or obesity. Women, infants, children, and adolescents are at particular risk of malnutrition, and poverty exacerbates the risk of malnutrition [51].

This study showed that those living in BU endemic regions in the Ashanti and Central regions of Ghana had a typical rural West African diet [31,52], which is mainly carbohydrate-based. The diets of Ghanaians have been undergoing an evolution over recent years, particularly in urban areas [53], where diets are shifting towards a 'Western-like' pattern [52,54]. However, the present study confirms that in rural areas, the dietary pattern has remained predominantly carbohydrate-based. Apart from socio-cultural beliefs [54], the accessibility and relatively cheap cost of starchy staples at the study sites may be the reason. Cereal-based meals like akple, kenkey, and rice and root tubers like yam, cassava, and cocoyam (used in preparation of staples like fufu and ampesi), were the most consumed by the study participants. Fufu is a well-known staple food in Ghana and is eaten by most ethnic groups and is prepared by

pounding cassava and plantain or yam/cocoyam, and is consumed with soup. Additionally, phytate content of plant-based diets will reduce absorption of micronutrients including zinc and iron.

Despite the high proportion of calories from carbohydrate, the estimated requirement was not met by most of the Cohort 1 participants. Other studies have also reported nutritional inadequacies in rural (but not urban) Ghana [31,50,55]. Moreover, the energy intake was lower than reported for people in Eastern and Southern Africa, which were 2245 and 2618 Kcal/capita/day, respectively [56]. This suggests that diets in BU endemic areas are poorer than African averages, highlighting the need for further research to understand nutrition in such marginalised communities. Indeed, this is the first study to report in the diets of people in these areas of Ghana.

Protein in the diets of the study participants was inadequate in about 50% of the participants. This was in line with a cross-sectional study that found low protein consumption more often in rural Ghana compared to people residing in urban areas and migrants in Europe [31]. Moreover, in a wildlife-depleted farm-forest landscape in South-Western Ghana, between 14% and 60% of households ate less than the recommended daily allowance of protein [57]. Although livestock rearing is common in the study sites, Cohort 1 participants frequently ate fish and poultry, and rarely ate red or processed meat. The cost of meat could be a factor affecting the low protein intake, with the majority of the protein coming from starchy staples, which may then reduce the intake of some essential amino acids.

In terms of micronutrients, Cohort 1 also had a high prevalence of dietary inadequacy of vitamins A, $B_{12}$, E, K and zinc. In line with this, overall micronutrient intake has been shown to be inadequate in other studies of diet in West Africa and Ghana. This is thought to be because they are mostly found in the vegetables, fruits, and nuts that are found in the soups, stews, and sauces that accompany the main starchy component of a meal [58–61]. While, some micronutrients can be found in carbohydrate-rich foods, including roots and tubers that were frequently consumed by Cohort 1 participants, their low DDS could be the source of this inadequacy. This is because it suggests that many of them ate from a relatively small palette of foods on a regular basis.

In the context of a population at risk of BU, the widespread inadequacy of micronutrient intake is interesting because they are essential for immune function [20] and are also seen in patients with chronic wounds. For example, inadequacies of serum vitamins A, E, zinc, and carotenes have been reported in patients with chronic leg ulcers [62]. During the inflammatory stage of wound healing, vitamin E helps reduce free radical damage caused by neutrophils [63]. On the other hand, vitamin K is necessary for the functionally essential post-translational modification of proteins involved in blood coagulation, which is known to be linked to the histopathology of the disease [8,64,65]. Inadequacy in vitamin K is known to cause haemorrhage and purpura in newborns and has recently been shown to affect bone and vascular health [66]. Zinc status has previously been reported to be low in Ghana [67] (prevalence of zinc deficiency ~22% in 2005 [68, 69]), and the prevalence of zinc deficiency in Sub-Saharan Africa varies between 15-50%. The current work found higher prevalence of zinc deficiency at 58%, further emphasising the poor quality of BU endemic diets compared to the country at large.

## Potential role for environmental enteric dysfunction (EED) in malnutrition in BU endemic areas

Despite finding no overt systemic inflammation in BU cases, we did find evidence for circulating cytokines above normal levels (low pg/ml concentrations) in a proportion (approximately 10-50%) of both cases and controls of Cohort 2. Indeed, this is in line with studies of healthy

controls in other low- and middle-income countries (LMICs), including Ghana [70–72]. Of particular note, elevated IL-6 levels were measured in more than half the participants. This likely reflects participants who were currently or recently co-infected with other pathogens. For instance, malaria infection can modulate circulating cytokines [73]. However, it could also reflect a prevalence of environmental enteric dysfunction (EED), a syndrome linked to areas of poor sanitation and hygiene that is characterised by weight loss, and histopathological evidence of chronic infection without a clearly diagnosed infection being present [74]. Notably it can cause stunting in children and here approximately one third of the young participants had stunted growth, adding further weight to this hypothesis. This is higher than national and regional prevalence, which in 2022 was reported to be 17% (20% in rural areas) and 17% in the Ashanti region in under 5 year olds, respectively [75]. One of the consequences of EED is malabsorption of nutrients (with the best evidence for zinc) [76], which could potentially affect the interpretation of our findings. On the other hand, IL-6 levels did not correlate inversely with those of zinc, suggesting that the degree of systemic inflammation in patients was not sufficient to cause the physiological and molecular changes that resulted in hypozincaemia [77]. Moreover, they were observed in the absence of a high level of CRP, which would be an indication of systemic inflammation [78]. Additional specific studies aimed at identifying EED, co-infections and systemic inflammation would be required to find out if this is a general feature of the population and whether this influences BU susceptibility and/or pathogenesis.

One interesting and unexpected finding was that, despite vitamin C having similar prevalence of dietary sufficiency (82%) and generally a good intake in their diets, many participants had serum vitamin C levels below the assumed normal range (55-88% depending on the cut-off used). This could potentially be attributed to these measurements being taken from two independent cohorts. However, both cohorts came from the same distribution of BU-endemic communities surrounding Agogo, Tepa and Dunkwa, ruling out local availability of food. Moreover, since BU disease has become less common, both cohorts were recruited over a long period (Cohort 1; July 2018 to May 2022, Cohort 2; October 2008 to June 2016), and therefore experienced similar seasonal variation in food availability. Therefore, it seems that even in the setting of adequate vitamin C intake, this did not translate to normal circulating vitamin C. This could be due to several factors, including that the main source of vitamin C is likely to be stews that may have been boiled for extensive periods, meaning that the intake was an over-estimate as the nutrient was destroyed [79]. It is also possible that participants over-estimated their intake during the 24-hour recall for psychological reasons or that EED could contribute since intestinal inflammation may be a confounding factor in vitamin C status [67,74] as it is absorbed in the jejunum and ileum.

## Insights into the role of diet in the pathogenesis of BU

While we could see no difference in the patterns of foods consumed between the BU cases and controls in the FFQ, the 24-hour recall showed that the cases consumed diets with a significantly lower DDS than the controls. This is in line with our findings that macronutrient intake (predominately carbohydrate) was similar between BU cases and controls, but the intake of micronutrients was different. In other words, it suggests that the lower dietary diversity of the BU patients could be one cause of the lower micronutrient intake due to the narrower range of foods eaten. In the context of a predominantly carbohydrate diet, a recent mouse study is of interest when considering underlying risk factors for BU in this population. Here, a ketogenic diet was modelled using b-hydroxybutyrate (the main ketone body produced in such diets), and this suppressed skin ulceration by stopping *M. ulcerans* growth and mycolactone synthesis with contrasting results observed in mice with a carbohydrate-rich diet [69]. Therefore, the

bacterium seems to require a diet rich in carbohydrates for growth and mycolactone production, which is more than adequately provided by the diet of the participants.

When considering micronutrient differences in detail, BU cases' diets contained significantly less selenium and vitamin $B_{12}$ and they were more likely to have dietary insufficiencies of these micronutrients. Vitamin $B_{12}$ deficiency is prevalent in regions with a high burden of NTDs including helminth infection, Chagas disease and leprosy [80]. However, vitamin $B_{12}$ was recently shown to be an important host-requirement for Mtb infection, as it has lost the biosynthetic machinery for this vitamin [81]. In contrast, selenium, a trace element, is required to clear mycobacteria [15]. In *M. tuberculosis* infection, selenium was shown to have a restricting function against MTB by regulating autophagy in macrophages contributing to low levels in patients compared to controls [82].

**Zinc.** Another of the largest differences in micronutrient intake between BU cases and controls was in zinc, which was 20% lower in cases. Here we were able to confirm, in an independent cohort, that serum levels were similarly 25% lower in cases than control. Moreover, BU patients were much more likely to have serum zinc levels below that of the assumed normal range than control counterparts. We are confident this represents a genuine micronutrient deficiency, since CRP levels were extremely low in both groups (below 0.05 mg/dL) so we did not need to correct this [83]. Moreover, our IL-6 data supports that both cases and controls had equal exposure to other infections and/or EED, and hence BU is associated with an additional reduction in zinc. Our findings relating to circulating cytokines are in line with previous reports of BU, which were found to be either unchanged or reduced compared to control groups [70,84]. This is thought to be due to the anti-inflammatory activity of mycolactone, which blocks the Sec61 translocon, causing a sharp reduction in the production of proinflammatory cytokines that are reliant on Sec61 for secretion [6]. Therefore, while it is difficult to separate cause vs. effect for micronutrient measurement, our finding that BU patients' nutritional intake of zinc was lower than controls supports a nutritional component to the susceptibility to disease.

For BU patients, inadequate zinc may be implicated in disease susceptibility and progress due to the role that this mineral plays particularly in the immune response. Indeed, zinc supplementation has been reported to be an effective solution for immune recovery and wound healing [85]. Zinc deficiency has been shown to decrease the production of TNF-α, IFN-γ and IL-2 by peripheral blood mononuclear cells [86]. Zinc deficiency is reported to increase susceptibility to leishmaniasis and TB [87], and in TB and leprosy hypozincaemia has been observed at the time of diagnosis [17]. In leprosy, there is a gradual reduction in serum zinc concentration as severity moves from Tuberculoid Leprosy to Lepromatous Leprosy, affecting Th1/ Th2 cytokine responses [88]. Furthermore, in humans, low serum zinc concentrations in cases with cutaneous leishmaniasis increase during antimonial therapy. This may be due to pre-existing zinc deficiencies or related to redistribution associated with the APR of visceral leishmaniasis [87,89].

**Vitamin C.** The findings for vitamin C are more difficult to interpret. When comparing BU cases with controls, intake of vitamin C was >30% higher in BU cases of Cohort 1 although this did not reach statistical significance. On the other hand, serum levels were 20% lower in Cohort 2, and a higher proportion of BU cases had abnormal serum levels that was highly significant. Furthermore, within the BU cases, there was also an association between BU lesion category and serum levels of vitamin C (in Cohort 2). Indeed, those with the more serious plaque form of the disease had significantly lower vitamin C serum levels than those with its earliest presentation, the nodule. Since all nodules are Category I lesions, it is notable that the few Category III lesions in Cohort 2 were all associated with vitamin C levels below the lower limit of the normal range. While these were approximately half those of the

Category I group, this difference did not reach statistical significance, perhaps due to the low numbers in this sub-group.

Therefore, while acknowledging that dietary and biomarker data were collected in different cohorts, our data suggests that the presence of *M. ulcerans* infection itself can affect vitamin C status. Notably, others have found this to be the case in patients with sepsis [90] and a cross-sectional analysis of the NHANES database found a negative correlation between inflammatory markers and vitamin C [39] possibly due to increased demands from the inflammatory response. Although vitamin C levels did not correlate with IL-6 in the current work, it may be that the presence of chronic, localised inflammation in BU patients is influencing the vitamin C levels in these patients.

Vitamin C is well-known to have important functions in skin health [91] as it is required for the production and stability of collagen [91,92]. Furthermore, a deficiency in vitamin C dysregulates both phagocytosis and the respiratory burst [93]. The stimulation of proinflammatory and Th1 cytokines is lower during vitamin C deficiency as it alters redox signalling, essential during differentiation of Th cells [94]. We speculate that mycolactone's known effect on extracellular collagen [8,95] could synergise with these lower vitamin C levels to have serious consequences for skin integrity.

## Limitations and conclusion

There are some limitations that should be noted in this study. Here, dietary patterns and dietary intake of the study group were assessed using methodological tools that rely on the memory of the subjects, which can be biased. It is known that participants tend to underestimate or overestimate dietary recalls, especially in food frequency questionnaires, even though efforts were made to reduce possible biases to the minimum. The study also includes two independent cohorts of BU patients and controls that were matched as closely as possible. In Cohort 2, anthropometry data was not collected when these were recruited, and so could not be reported It was not possible to determine the inflammation markers in all patients, and our attempt to measure vitamin $B_{12}$ in this same cohort was subject to assay failure.

Despite these limitations, these findings align with two recent independent reports of BU patients in Côte d'Ivoire [96,97]. While there was no information on controls, in one study, 51 BU patients also ate predominantly starchy foods and minimal proteins; indeed, dairy products, fruits, vegetables, and vegetable oils were consumed by only 8% of the cases [97]. Their food choices were found to be influenced by societal norms where food types were prohibited in their communities, including certain animal and plant proteins, vegetables, and fruits. Being underweight tended to be higher in Côte d'Ivoire vs Ghanaian BU cases (33-53% vs 19-23%), and acute malnutrition was also reported [97]. Interestingly, CRP levels >20 mg/L were reported in 7-20% of the cases, suggesting that EED may be more prevalent in Côte d'Ivoire. However, in a subset of these patients and similar to here, zinc was found to be below normal levels in all BU patients [96]. Importantly, nutritional assistance provided at one health centre significantly improved wound healing alongside marked improvement in anthropometrics and anaemia compared to another where no nutritional support was included in BU patients' management package [97]. Taken together, the findings strongly support large scale randomised control studies for nutritional intervention in at-risk communities for BU and nutritional support for all BU patients, alongside a detailed assessment of the prevalence of EED and how this may influence susceptibility to BU and other NTDs.

## Supporting information

**S1 Fig. The most and least frequently consumed food groups by the study participants.** A food frequency questionnaire was put to all 80 Cohort 1 participants (40 BU cases, 40 controls), covering a total of 105 foods from 25 food groups. There are 4 possible answers for

frequency of consumption for each food, namely daily, weekly, monthly, and rarely/never. This figure visualises the distribution of the most frequent consumption of one or more foods in that food group in the past month amongst the participants. The red line represents the frequency of consumption, and its length is proportional to the number of participants with that maximal frequency. Data from the food groups showed that at least 60% of the participants ate foods from these food groups at least weekly, where similar numbers consumed foods monthly, rarely or never vs. weekly or daily (N-Q), and those where at least 60% of the participants ate them monthly, rarely or never (R-Y).
(TIF)

**S2 Fig. No correlation between nutrition and inflammation markers in BU cases and controls.** Serum concentrations of vitamin C, zinc and IL-6 were analysed using Spearman's correlation, and are presented as a heatmap. Spearman's correlation coefficients are given along with P-value (in brackets).
(TIF)

**S3 Fig. No correlation between nutrition and inflammation markers in BU cases and controls.** Serum concentrations of vitamin C, zinc and IL-6 were analysed using Spearman's correlation, and are presented as a heatmap. Spearman's correlation coefficients are given along with P-value (in brackets).
(TIF)

**S1 Table. List of foods used in the FFQ administration.**
(DOCX)

**S2 Table. Food groups used in the Dietary Diversity Score, generated from 24-hour recall data.**
(DOCX)

**S3 Table. Socioeconomic status of Cohort 1 study participants.** a; T $\chi^2$ test. p-value < 0.05 was statistically significant.
(DOCX)

**S4 Table. Mean percentage of macronutrients in the diet of Cohort 1 participants.** Unit conversions: 1g of carbohydrate is equivalent to 4 Kcal, 1g of protein is equivalent to 4 Kcal and 1g of fats is equivalent to 9 Kcal, which were then used to calculate percentages.
(DOCX)

**S5 Table. Comparison of energy and nutrient intake of Cohort 1 study participants between sexes.** Data were compared in cases and controls between males and females using a Mann-Whitney test. P-value <0.05 are indicated in bold text.
(DOCX)

**S6 Table: Comparison of energy and nutrient intake of Cohort 1 study participants between those with different household income.** Data were compared in cases and controls between those with different levels of household monthly income, using the Kruskal-Wallis test and Dunn's post-hoc test for multiple comparisons.
(DOCX)

## Acknowledgement

The authors would like to thank all study participants and community leaders, as well as the nurses, diseases control officers, and physiotherapists from Agogo Presbyterian Hospital, Tepa Government Hospital and Dunkwa Government Hospital who support BU patient

management. We are grateful to the District Directors and the National Buruli ulcer Control programme. We thank Dr Peter Williams (University of Surrey) for statistical advice.

## Author contributions

**Conceptualization:** Aloysius Dzigbordi Loglo, Richard Odame Phillips, Reginald Annan, Barbara Engel, Rachel E Simmonds.

**Formal analysis:** Aloysius Dzigbordi Loglo, Rachel E Simmonds.

**Funding acquisition:** Aloysius Dzigbordi Loglo, Richard Odame Phillips, Reginald Annan, Rachel E Simmonds.

**Investigation:** Aloysius Dzigbordi Loglo, Philemon Boasiako Antwi, Kabiru Mohammed Abass, Samuel Osei-Mireku, George Amofa, Elizabeth Ofori, Jonathan Kofi Adjei, Michael Ntiamoah Oppong.

**Methodology:** Aloysius Dzigbordi Loglo, Rachel E Simmonds.

**Project administration:** Aloysius Dzigbordi Loglo, Richard Odame Phillips, Reginald Annan, Barbara Engel, Rachel E Simmonds.

**Resources:** Aloysius Dzigbordi Loglo, Richard Odame Phillips, Rachel E Simmonds.

**Supervision:** Richard Odame Phillips, Reginald Annan, Barbara Engel, Rachel E Simmonds.

**Validation:** Aloysius Dzigbordi Loglo, Reginald Annan, Barbara Engel, Rachel E Simmonds.

**Visualization:** Aloysius Dzigbordi Loglo, Rachel E Simmonds.

**Writing – original draft:** Aloysius Dzigbordi Loglo, Rachel E Simmonds.

**Writing – review & editing:** Aloysius Dzigbordi Loglo, Richard Odame Phillips, Reginald Annan, Barbara Engel, Rachel E Simmonds.

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
