## [Decision Letter · Decision Letter 0]

25 Jan 2025

Dear Prof. Simmonds,

We are pleased to inform you that your manuscript 'Micronutrient-deficient diets and possible environmental enteric dysfunction in Buruli ulcer endemic communities in Ghana: lower dietary diversity and reduced serum zinc and vitamin C implicate micronutrient status a possible susceptibility factor' has been provisionally accepted for publication in PLOS Neglected Tropical Diseases.

Best regards,

Joseph M. Vinetz

Section Editor

Joseph Vinetz

Section Editor

Shaden Kamhawi

co-Editor-in-Chief

Paul Brindley

co-Editor-in-Chief

Reviewer's Responses to Questions

**Key Review Criteria Required for Acceptance?**

**Methods**

-Are the objectives of the study clearly articulated with a clear testable hypothesis stated?

-Is the study design appropriate to address the stated objectives?

-Is the population clearly described and appropriate for the hypothesis being tested?

-Is the sample size sufficient to ensure adequate power to address the hypothesis being tested?

-Were correct statistical analysis used to support conclusions?

-Are there concerns about ethical or regulatory requirements being met?

Reviewer #1: The organization and analysis of the survey, immunological, and demographic data are appropriate and clearly described. The authors clearly present the study design, the populations studied, and all survey vehicles used.

**Results**

-Does the analysis presented match the analysis plan?

-Are the results clearly and completely presented?

-Are the figures (Tables, Images) of sufficient quality for clarity?

Reviewer #1: Tables and figures are clearly organized with appropriate statistical analysis. A variety of data presentations are used to best illustrate conclusions from the results.

**Conclusions**

-Are the conclusions supported by the data presented?

-Are the limitations of analysis clearly described?

-Do the authors discuss how these data can be helpful to advance our understanding of the topic under study?

-Is public health relevance addressed?

Reviewer #1: The conclusions are clearly presented with interesting local and global implications. The limitations are discussed throughout the article and then summarized at the end of the article. The public health applications for BU treatment and the overall health of the rural Ghanan communities are clearly discussed.

**Editorial and Data Presentation Modifications?**

Reviewer #1: The only modification recommended in a final careful grammatical review to remove a few minor issues. Otherwise the article is ready for publication.

**Summary and General Comments**

Reviewer #1: In this article Loglo et al describe the results of a long-term study on the correlation between diet and the prevalence of Buruli ulcer (BU) caused by infection from Mycobacterium ulcerans. Using a combination of blood sample analysis, food questionnaires, and demographic data, the authors measure the connection between diet, micro- and macronutrient availability, immune markers, and BU infection within multiple rural communities in Ghana. Given the difficulty in obtaining matching patients with and without BU and the low overall prevalence of BU, this study combines two separate cohorts over ~16 years and has low numbers of patients withing certain subgroups, limiting the statistical power of the analysis. The authors are open and clear about this limitation and other limitations of the study (dependence on food surveys and lack of matching food data for cohort 2). This openness strengthens the other conclusions from the article. Overall, interesting conclusions are made about the link between BU and micronutrient deficiencies with zinc and vitamin C and a possible link to environmental enteric dysfunction. General conclusions about the high carbohydrate, lower protein diet prevalent in rural Ghana and its impact on population growth and proper immune function provide interesting applications for public health policy and global food programs. The article could use a final grammatical edit but otherwise is prepared for publication in PLOS Neglected Tropical Diseases without further scientific edits.

PLOS authors have the option to publish the peer review history of their article (what does this mean?). If published, this will include your full peer review and any attached files.

Reviewer #1: No

---

## [Editor Report · Acceptance letter]

Dear Prof. Simmonds,

We are delighted to inform you that your manuscript, "Micronutrient-deficient diets and possible environmental enteric dysfunction in Buruli ulcer endemic communities in Ghana: lower dietary diversity and reduced serum zinc and vitamin C implicate micronutrient status a possible susceptibility factor," has been formally accepted for publication in PLOS Neglected Tropical Diseases.

Best regards,

Shaden Kamhawi

co-Editor-in-Chief

Paul Brindley

co-Editor-in-Chief
